# Quantitative analysis of chest computed tomography of COVID-19 pneumonia using a software widely used in Japan

**Minako Suzuki**[1]*, **Yoshimi Fujii**[2], **Yurie Nishimura**[2], **Kazuma Yasui**[2],
**Hidefumi Fujisawa**[1]

1 Department of Radiology, Showa University Northern Yokohama Hospital, Yokohama, Kanagawa, Japan,
2 Department of Radiology, Fujisawa City Hospital, Fujisawa, Kanagawa, Japan

☯ These authors contributed equally to this work.
* minacom69@gmail.com

## Abstract

This study aimed to determine the optimal conditions to measure the percentage of the area considered as pneumonia (pneumonia volume ratio [PVR]) and the computed tomography (CT) score due to coronavirus disease 2019 (COVID-19) using the Ziostation2 image analysis software (Z2; Ziosoft, Tokyo, Japan), which is popular in Japan, and to evaluate its usefulness for assessing the clinical severity. We included 53 patients (41 men and 12 women, mean age: 61.3 years) diagnosed with COVID-19 using polymerase chain reaction who had undergone chest CT and were hospitalized between January 2020 and January 2021. Based on the COVID-19 infection severity, the patients were classified as mild (n = 38) or severe (n = 15). For 10 randomly selected samples, the PVR and CT scores by Z2 under different conditions and the visual simple PVR and CT scores were compared. The conditions with the highest statistical agreement were determined. The usefulness of the clinical severity assessment based on the PVR and CT scores using Z2 under the determined conditions was statistically evaluated. The best agreement with the visual measurement was achieved by the Z2 measurement condition of $\geq$–600 HU. The areas under the receiver operating characteristic curves, Youden's index, and the sensitivity, specificity, and p-values of the PVR and CT scores by Z2 were as follows: PVR: 0.881, 18.69, 66.7, 94.7, and <0.001; CT score: 0.77, 7.5, 40, 74, and 0.002, respectively. We determined the optimal condition for assessing the PVR of COVID-19 pneumonia using Z2 and demonstrated that the AUC of the PVR was higher than that of CT scores in the assessment of clinical severity. The introduction of new technologies is time-consuming and expensive; our method has high clinical utility and can be promptly used in any facility where Z2 has been introduced.

## Introduction

The coronavirus disease 2019 (COVID-19) pandemic caused by the novel coronavirus severe acute respiratory syndrome coronavirus 2 (SARS-CoV-2) was first identified in Wuhan,

**Funding:** The authors received no specific funding for this work.

**Competing interests:** The authors have declared that no competing interests exist.

China, and reported in December 2019 [1]. The pandemic prevails with an increasing number of infections and deaths. In Japan, the first COVID-19 case was reported in January 2020 [2], and by April 2023, over 33 million people had been infected, and more than 74000 people had died [3]. After the Omicron strain of SARS-CoV-2 became prevalent, the number of severe cases complicated by pneumonia decreased, and the vaccination spread socially. In May 2023, the legal classification was changed, and it was decided that COVID-19 would be treated on the same level as an influenza virus infection [3].

During the study period, COVID-19 had a high complication rate with pneumonia, especially in older adults [4,5], with a high rate of aggravation and mortality, and it became necessary to distribute limited medical resources. The discrimination between mild and severe cases at the emergency department was an important and burdensome task. Typically, the severity was determined by symptoms, age, complications, blood tests, and computed tomography (CT) findings. The CT findings were generally evaluated visually, and CT scores based on visual evaluation were not accurate or objective and took time and effort on the part of the evaluator. There are many reports on the CT severity assessment of COVID-19-associated pneumonia using imaging software. The measurement methods and evaluation conditions differ for each individual tool, and few of them have been widely adopted in clinical settings.

The Ziostation2 image analysis software (Z2; Ziosoft, Tokyo, Japan) had been introduced in approximately 300 facilities in Japan and was designed to quantify pulmonary emphysema in patients with chronic obstructive pulmonary disease. The Z2 software was released in 2010, and it can automatically extract the bronchi, lungs, and low-attenuation area (LAA). Moreover, it can measure the volume of the entire lung and the LAA in cases of chronic obstructive pulmonary disease. When a region above a certain concentration is recognized as a pneumonia region, the pneumonia volume ratio (PVR) can be measured by changing the threshold setting of the CT value (Fig 1A and 1B). To the best of our knowledge, there are no reports of COVID-19 pneumonia assessment by Z2 to date.

As Z2 has not been set to evaluate pneumonia, it is necessary to determine the threshold in Hounsfield units (HUs) for it. Therefore, we decided to set the threshold at the concentration that most closely matched the visual evaluation.

This study was conducted to determine how the radiology department of a city hospital in Japan could use existing image analysis software to contribute to clinical practice at a time when the pre-Delta strain COVID-19 virus was predominant. Therefore, in this study, we determined the appropriate conditions for the evaluation of COVID-19 pneumonia by Z2 through comparison with visual evaluation results and examined the usefulness of the clinical severity assessment of Z2.

## Materials and methods

### Study population

This study adhered to the principles of the Declaration of Helsinki and was approved by the Ethical Review Committee of Fujisawa City Hospital (approval number: F2021022). The study was conducted retrospectively using imaging data and electronic medical records. An informed consent was provided by all patients in an opt-out manner on the website. The initiation of the study period was from the date of institutional ethics committee approval (October 19, 2021) to March 31, 2022.

We evaluated patients diagnosed with COVID-19 using a polymerase chain reaction test who required a chest CT scan at our hospital and inpatient hospital care between January 2020 and January 2021. The patients who had an initial CT scan at another hospital or those who were initially treated at another hospital and subsequently transferred to our hospital, and

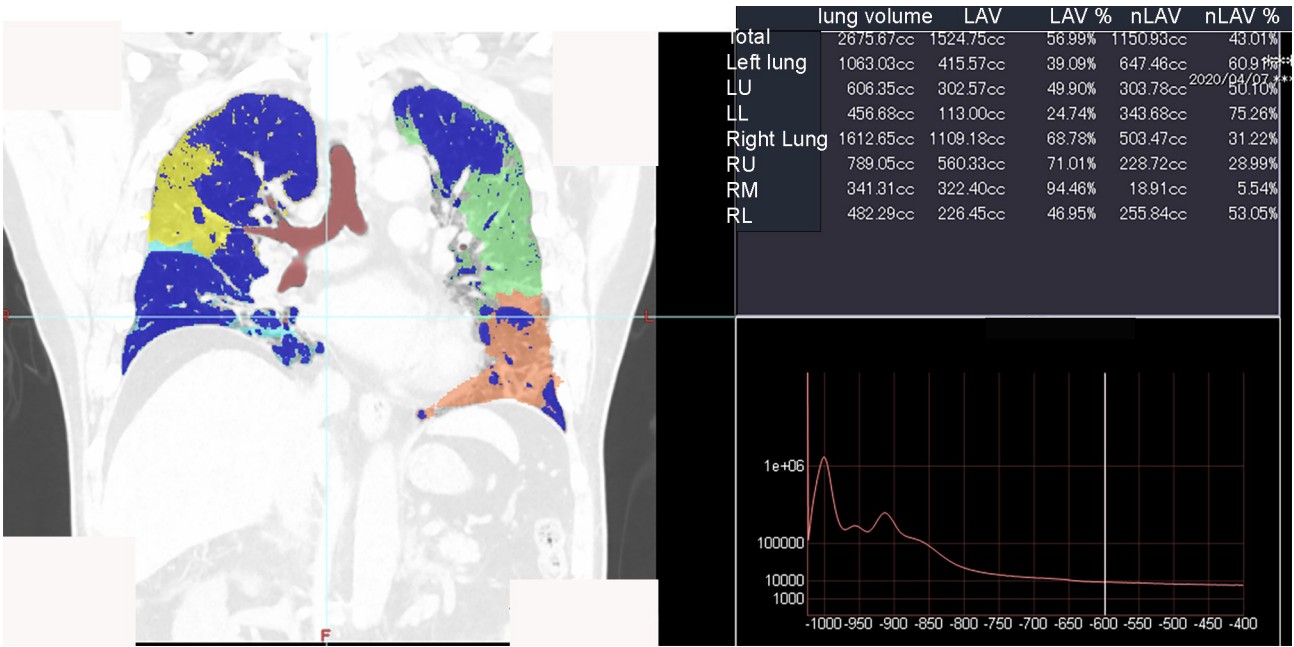

| | lung volume | LAV | LAV % | nLAV | nLAV % |
|---|---|---|---|---|---|
| Total | 2675.67cc | 1524.75cc | 56.99% | 1150.93cc | 43.01% |
| Left lung | 1063.03cc | 415.57cc | 39.09% | 647.46cc | 60.91% |
| LU | 606.35cc | 302.57cc | 49.90% | 303.78cc | 50.10% |
| LL | 456.68cc | 113.00cc | 24.74% | 343.68cc | 75.26% |
| Right Lung | 1612.65cc | 1109.18cc | 68.78% | 503.47cc | 31.22% |
| RU | 789.05cc | 560.33cc | 71.01% | 228.72cc | 28.99% |
| RM | 341.31cc | 322.40cc | 94.46% | 18.91cc | 5.54% |
| RL | 482.29cc | 226.45cc | 46.95% | 255.84cc | 53.05% |

**Fig 1. Images displayed on the console of the Z2.** Z2 monitor screen. The PVR above a certain concentration is displayed in the upper right corner (red square). LAV, low attenuation volume; LL, left lower lobe; LU, left upper lobe; nLAV, not LAV (lung volume other than LAV); PVR, pneumonia volume ratio; RL, right lower lobe; RM, right middle lobe; RU, right upper lobe; Z2, Ziostation2.

cases without pneumonia findings on chest CT were excluded. Ten samples were randomly selected from patients aged <65 years with an uncomplicated condition.

The clinical severity of COVID-19 was classified as mild (SpO$_2$ >93%) or severe (SpO$_2$ ≤93%, intubation, and intensive care unit management) based on the symptoms at the time of hospitalization, according to the guidelines of the Ministry of Health and Welfare [4]. The clinical severity, symptoms, comorbidities, blood test values, and clinical course were retrieved from the electronic medical records.

## CT protocol

The chest CT scans were obtained using 64-multidetector CT scanners (SOMATOM Definition AS 64; Siemens Healthineers, Erlangen, Germany). The CT parameters used at our hospital were as follows: 120 kVp, 160–316 mA current intelligent control (auto mA), and 5-mm slice thickness reconstruction. All CT examinations were performed without the use of intravenous contrast agents. The EV Report picture archiving and communication system (PACS) (PSP Corporation, Tokyo, Japan) was used to evaluate the CT findings.

## CT image analysis

Two radiologists evaluated the CT findings of pneumonia in all patients (Y.N. and M.S.) using both 5-mm images and high-resolution CT, in consultation for the presence or absence of ground-glass opacity (GGO) (–/+), crazy-paving finding (–/+), consolidation (none/mild/moderate/severe), and emphysema (–/+).

For the 10 selected participants, visual evaluation of the PVR was performed independently by two radiologists (Y.F. and M.S.) using the free-form curve drawing tool of the PACS by adding up the area of the lungs and the pneumonia area freehand at 1.5-cm intervals in the coronal chest CT images (Fig 2). In the same participants, the two radiologists independently scored

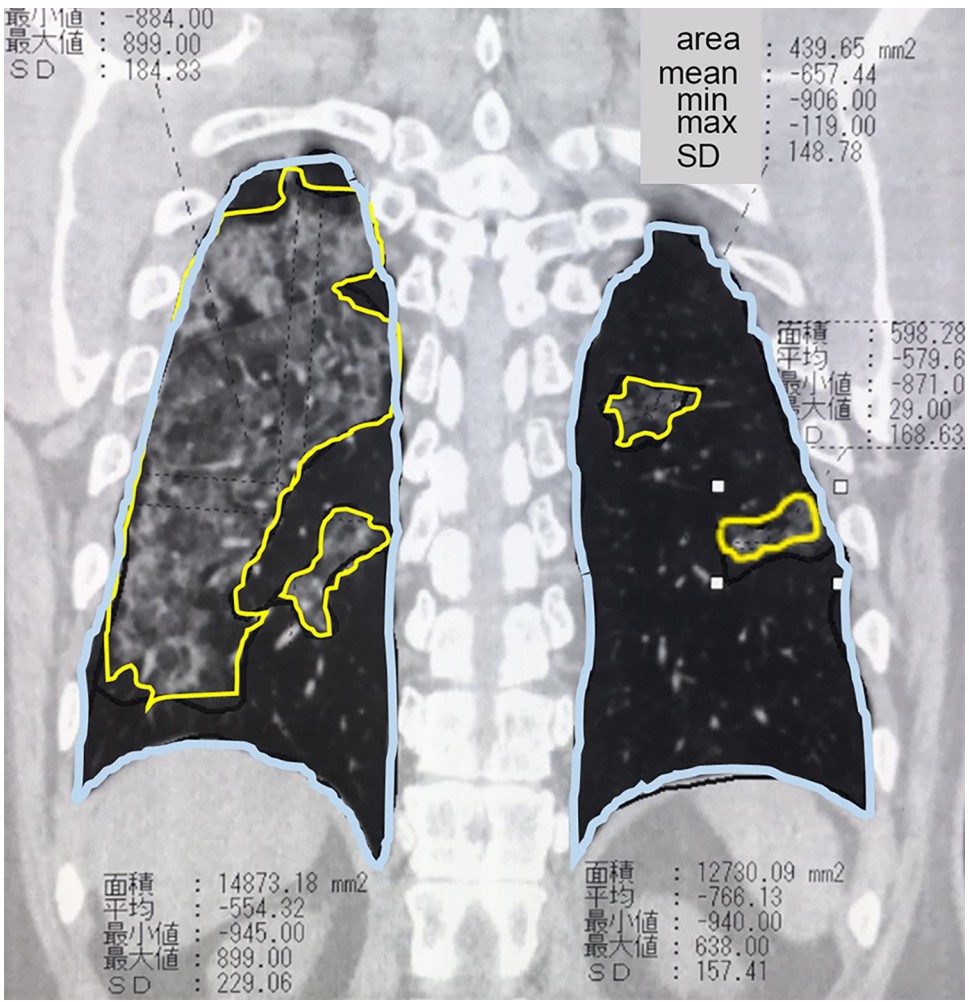

**Fig 2. Visual measurement of the PVR.** Two radiologists independently selected the entire lung field and pneumonia area every 1.5 cm on the coronal view using a drawing tool on the PACS (PSP Corporation, Tokyo, Japan) and added these up to measure the PVR. The blue line indicates the entire lung field ($mm^2$), and the yellow line indicates the pneumonia area ($mm^2$). The minimum and maximum in the figure represent CT values in the region. CT, computed tomography; max, maximum; min, minimum; PACS, report picture archiving and communication system; PVR, pneumonia volume ratio; SD, standard deviation.

the percentage of pneumonia area in each lobe using visual measurements (0: 0%, 1: 25%, 2: 25–50%, 3: 50–75%, and 4: 75–100%).

Z2 provided the quantification of the emphysema, healthy lung parenchyma, GGO, and consolidation based on a HU. Z2 can divide segments and calculate the total volumes for both the right and left lungs. In the measurement of the PVR and CT scores in the 10 selected participants using Z2, the lung fields above a particular concentration were set as pneumonia areas and measured at $\geq-500$ HU, $\geq-550$ HU, $\geq-600$ HU, $\geq-650$ HU, and $\geq-700$ HU. Z2 may not recognize the subpleural consolidation area as a lung field, and the total lung volume may be underestimated (Fig 3); therefore, radiologist A (M.S.) made the appropriate corrections manually.

## Statistical analysis

The presence of significant differences in participant background (age, sex, number of days from disease onset to CT evaluation, and laboratory test results) between the mild and severe

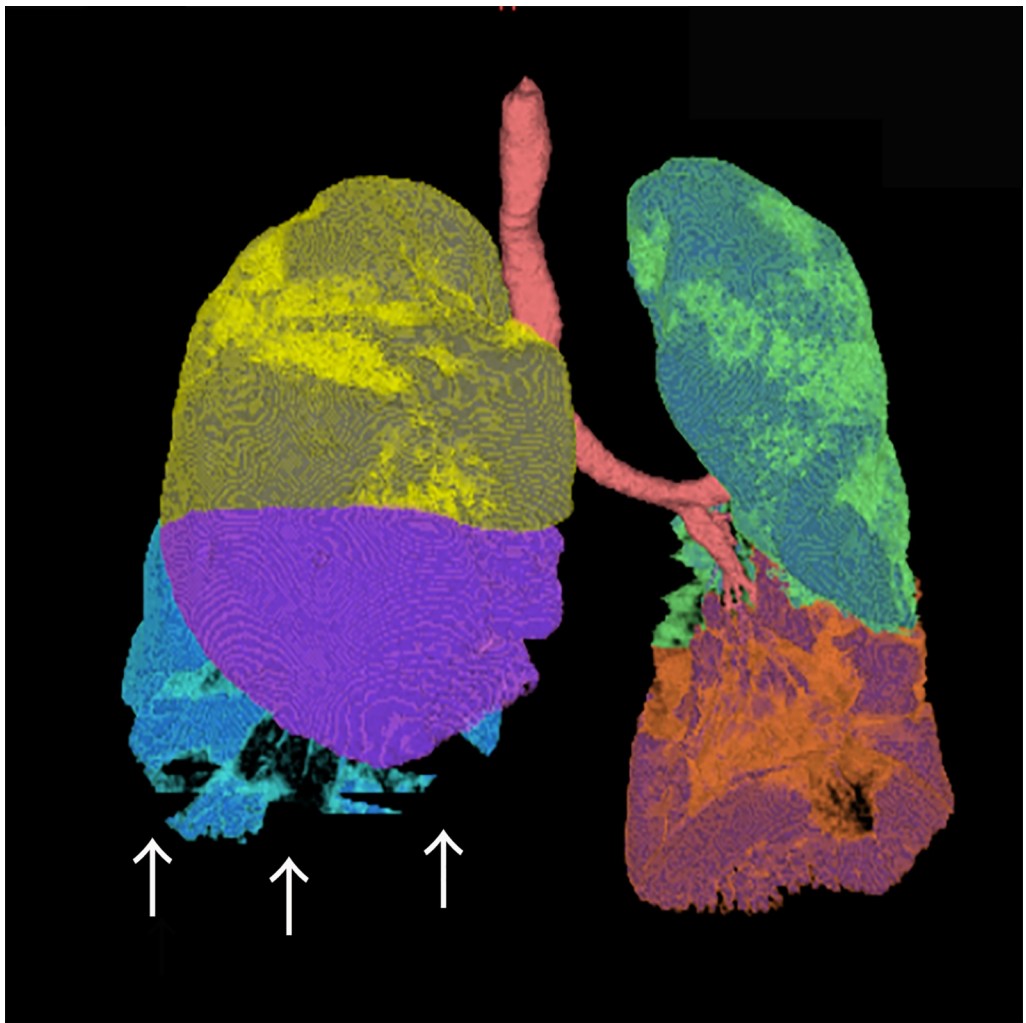

**Fig 3. Dorsal subpleural consolidations are not recognized as part of the lung and require manual correction.** The white arrows indicate the areas that need to be manually corrected.

groups was evaluated using the *t*-test and chi-square test. The accuracy between the gross measurements of the PVR and CT scores by two independent radiologists and the measurements by Z2 were evaluated using Spearman's rank correlation coefficient. The influence of possible confounding factors of participant background (age, sex, number of days from disease onset to CT evaluation, and presence of comorbidities) on the severity classification of the PVR by Z2 was evaluated using the bivariable logistic regression. The usefulness of the PVR and CT scores by Z2 under the determined conditions, primary laboratory tests, and CT findings in the clinical severity assessment was determined by the receiver operating characteristic (ROC) curves, Youden's index, sensitivity, specificity, and p-values. All statistical analyses were performed using the SPSS software (version 27; IBM, Armonk, NY, USA).

## Results

In total, there were 91 patients diagnosed with COVID-19 using a polymerase chain reaction test who required a chest CT scan at our hospital and inpatient hospital care between January 2020 and January 2021. Of these, three patients who received initial treatment at another

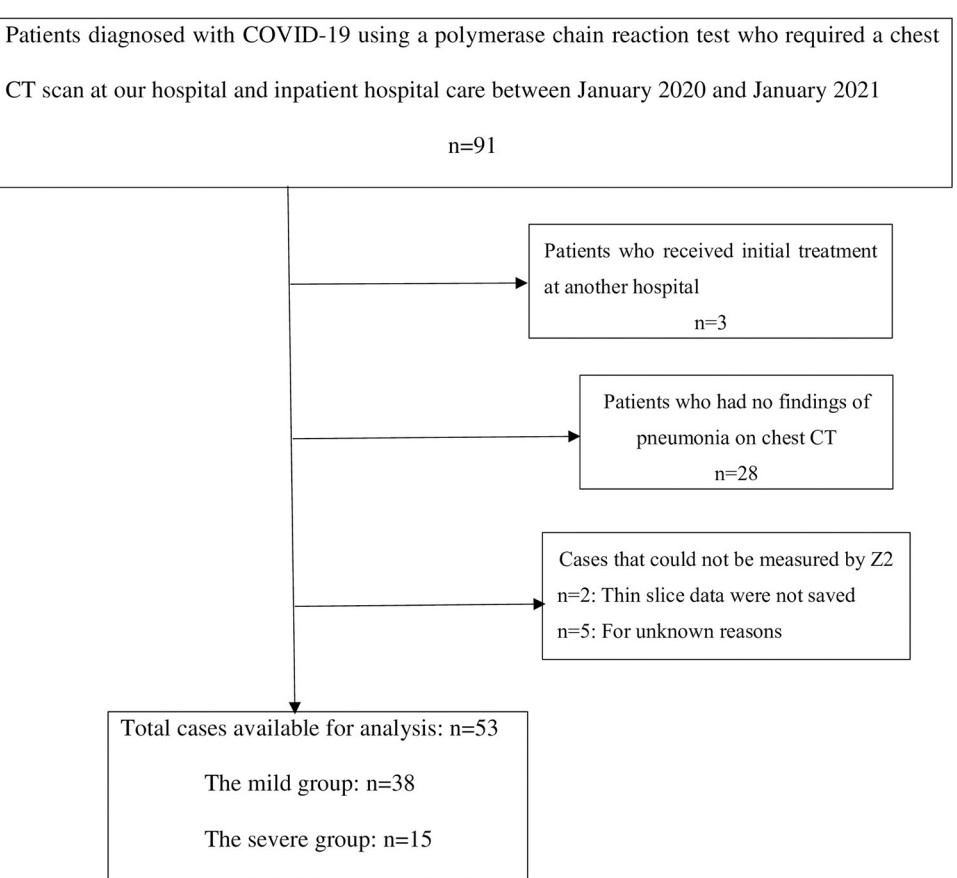

**Fig 4. The flow chart shows the process of determining the number of study cases to 53.**

hospital and 28 patients who had no findings of pneumonia on chest CT were excluded. Two cases were excluded from the study because the thin slice data necessary for Z2 measurement were not saved, and five cases could not be measured by Z2 for unknown reasons (Fig 4).

In total, 53 participants (41 men and 12 women, with a median age of 61.3 years; 38 in the mild group and 15 in the severe group) were included. Table 1 shows the participants' demographics (age, sex, and presence of comorbidities), laboratory findings, and CT findings. Fifty-two participants presented with COVID-19 symptoms; however, there was no significant difference in the severity of the symptoms between the mild and severe disease groups. Significant differences in the number of days from disease onset to CT evaluation and the presence of comorbidities were observed between the two groups. In addition, laboratory results revealed that the C-reactive protein (CRP) and lactate dehydrogenase (LDH) levels differed significantly between the two groups. The CT findings showed a significant difference in consolidation between the two groups.

Table 2 shows the results of Spearman's correlation between Z2 (under each condition; PVR: $\geq$−500 HU, $\geq$−550 HU, $\geq$−600 HU, $\geq$−650 HU, and $\geq$−700 HU, CT score: $\geq$−500 HU and $\geq$−600 HU) and the two radiologists for the PVR and CT scores in the 10 participants without comorbidities, respectively. While the accuracy between the two radiologists and Z2 for the PVR was equally high at $\geq$−500 HU to $\geq$−600 HU, the accuracy for CT scores was higher at $\geq$−600 HU than at $\geq$−500 HU. Based on these results, the Z2 measurement condition for COVID-19 pneumonia that achieved the best accuracy with the gross measurement was determined to be $\geq$−600 HU.

**Table 1. Patient background, blood test, and CT findings.**

| Factor | Total (n = 53) | Mild (n = 38) | Severe (n = 15) | p-value |
|---|---|---|---|---|
| Age (years; median) | 61.28 (66) | 58.95 (59.5) | 67.2 (67.00) | 0.148 |
| Sex (male; %) | 41 (77.4) | 27 (71.1) | 14 (93.3) | 0.081 |
| Date from onset to CT (range) | 6.0 (1–14) | 5.2 (1–12) | 7.9 (4–14) | 0.016 |
| Comorbidities[a] (%) | 18 (34.0) | 9 (23.7) | 9 (60.0) | 0.012 |
| DM (%) | 6 (11.3) | 3 (7.9) | 3 (20.0) | 0.21 |
| COPD (%) | 5 (9.4) | 2 (5.3) | 3 (20.0) | 0.098 |
| CRF (%) | 4 (7.5) | 3 (7.9) | 1 (6.7) | 0.879 |
| Obesity (%) | 6 (11.3) | 1 (2.6) | 5 (33.3) | 0.0015 |
| Malignancy (%) | 3 (5.7) | 1 (2.6) | 2 (13.3) | 0.129 |
| Symptoms (any) | 52 (98.1) | 37 (97.4) | 15 (100) | 0.526 |
| Fever (%) | 46 (86.8) | 33 (86.8) | 13 (86.7) | 0.986 |
| Cough (%) | 22 (41.5) | 14 (36.8) | 8 (53.3) | 0.272 |
| Taste disorder (%) | 7 (16.2) | 7 (18.4) | 0 (0) | 0.074 |
| Vomiting or diarrhea (%) | 9 (17.0) | 8 (21.1) | 1 (6.7) | 0.21 |
| Blood tests (range) | | | | |
| WBC (×10⁹/L) | 6.6 (2.5–21.7) | 6.3 (2.5–21.7) | 7.2 (3.5–12.3) | 0.33 |
| Lymphocytes (%) | 17.8 (4–46.5) | 11.6 (4–46.5) | 7.5 (5.2–26.4) | 0.085 |
| CRP (mg/dL) | 8.1 (0.02–29.6) | 2.9 (0.02–29.6) | 13.1 (2.4–26.0) | 0.005 |
| LDH (U/L) | 364.8 (144–1136) | 302.8 (144–834) | 521.8 (233–1136) | <0.001 |
| AST (U/L) | 50.2 (14–160) | 46.3 (14–160) | 60.1 (25–131) | 0.213 |
| ALT (U/L) | 43.5 (7–200) | 39.97 (7–200) | 52.27 (15–163) | 0.308 |
| Creatinine (mg/dL) | 1.39 (0.31–14.9) | 1.15 (0.31–14.9) | 0.998 (0.52–2.34) | 0.455 |
| eGFR (mL/min) | 65.8 (30–144) | 65.4 (1.6–144) | 66.93 (22–100) | 0.853 |
| CT findings | | | | |
| GGO (+) (%) | 53 (100) | 38 (100) | 15 (100) | 1 |
| Crazy paving (+) (%) | 8 (15.1) | 5 (13.2) | 3 (20.0) | 0.53 |
| Consolidation (–) (%) | 23 (43.4) | 21 (55.3) | 2 (13.3) | |
| (+) (%) | 17 (32.1) | 12 (31.6) | 5 (33.3) | |
| (++) (%) | 8 (15.1) | 5 (13.2) | 3 (20.0) | |
| (+++) (%) | 5 (9.4) | 0 (0) | 5 (33.3) | 0.0006 |
| Z2 (≥–600 HU) | | | | |
| PVR mean (median, range) | 12.44 (1.63–63.26) | 7.59 (1.63–40.11) | 24.71 (3.32–63.26) | <0.001 |
| CT score mean (median, range) | 6.62 (8, 5–15) | 5.87 (5, 5–11) | 8.53 (7, 5–15) | <0.001 |

ALT, alanine aminotransferase; AST, aspartate aminotransferase; COPD, chronic obstructive pulmonary disease; CRF, chronic renal failure; CRP, C-reactive protein; CT, computed tomography; DM, diabetes mellitus; eGFR, estimated glomerular filtration rate; GGO, ground-glass opacity; LDH, lactate dehydrogenase; PVR, pneumonia volume ratio; WBC, white blood cell; Z2, Ziostation2.

[a]Comorbidities were defined as the presence of any of the following: DM, COPD, severe cardiovascular disease, severe CRF, obesity, malignancy under treatment, immunosuppression, and liver cirrhosis.

Figs 5 and 6 show the ROC curves and boxplots corresponding to the classification of disease severity by the PVR and CT scores using Z2 (≥–600HU), CRP, and LDH. The areas under the curve (AUCs) were 0.881, 0.77, 0.788, and 0.842, respectively.

Youden's index values for the PVR and CT scores at ≥–600 HU by Z2, CRP, and LDH were 18.69, 7.5, 5.26, and 306.5, respectively. The sensitivities for the PVR and CT scores at ≥–600 HU by Z2 were 66.7% and 40%, respectively. The specificities for the PVR and CT scores at ≥–600 HU by Z2 were 94.7% and 74%, respectively. The p-value for the PVR at ≥–600 HU

**Table 2. Results of the Spearman's test of the PVR and CT scores by two radiologists and Ziostation2 of five/two conditions in the 10 selected patients.**

| Correlation coefficient | | Reader B | ≥–500 HU | ≥–550 HU | ≥–600 HU | ≥–650 HU | ≥–700 HU |
|---|---|---|---|---|---|---|---|
| Reader A | PVR | 0.879 | 0.976 | 0.976 | 0.976 | 0.964 | 0.818 |
| | CT score | 0.976 | 0.639 | | 0.651 | | |
| Reader B | PVR | | 0.842 | 0.842 | 0.842 | 0.83 | 0.661 |
| | CT score | | 0.584 | | 0.696 | | |

CT, computed tomography; PVR, pneumonia volume ratio.

Reader A: M.S., Reader B: Y.F.

by Z2 was p<0.001, and that for CT scores at ≥– 600 HU by Z2 was p = 0.002 (Table 3). The bivariable logistic regression of the PVR (≥–600 HU) according to age, sex, date from onset to CT, and comorbidities showed no significant effects, except for comorbidities (Table 4). The

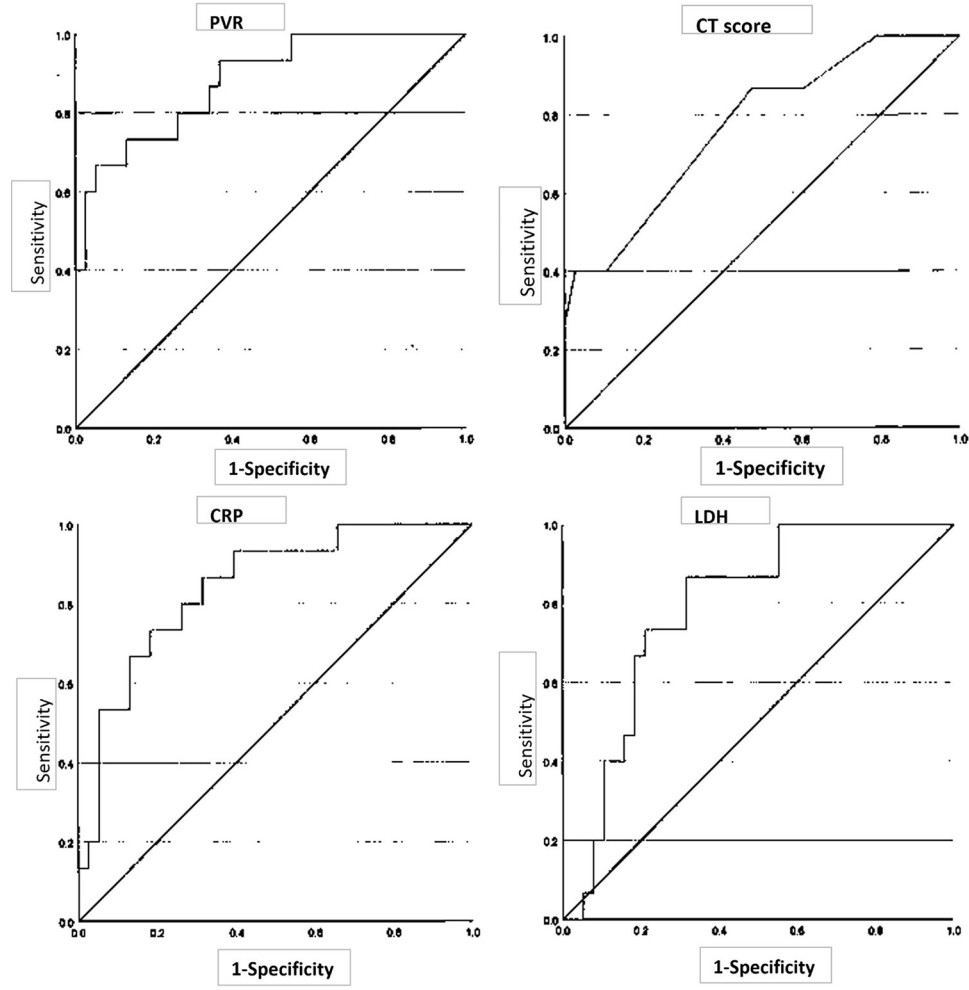

**Fig 5. ROC curves for the PVR, CT score, CRP, and LDH.** ROC curve for **a**. PVR using Z2 (≥–600 HU) and **b**. CT scores using Z2 (≥–600 HU), **c**. CRP, and **d**. LDH. CRP, C-reactive protein; CT, computed tomography; LDH, lactate dehydrogenase; PVR, pneumonia volume ratio; ROC, receiver operating characteristic; SD, standard deviation; Z2, Ziostation2.

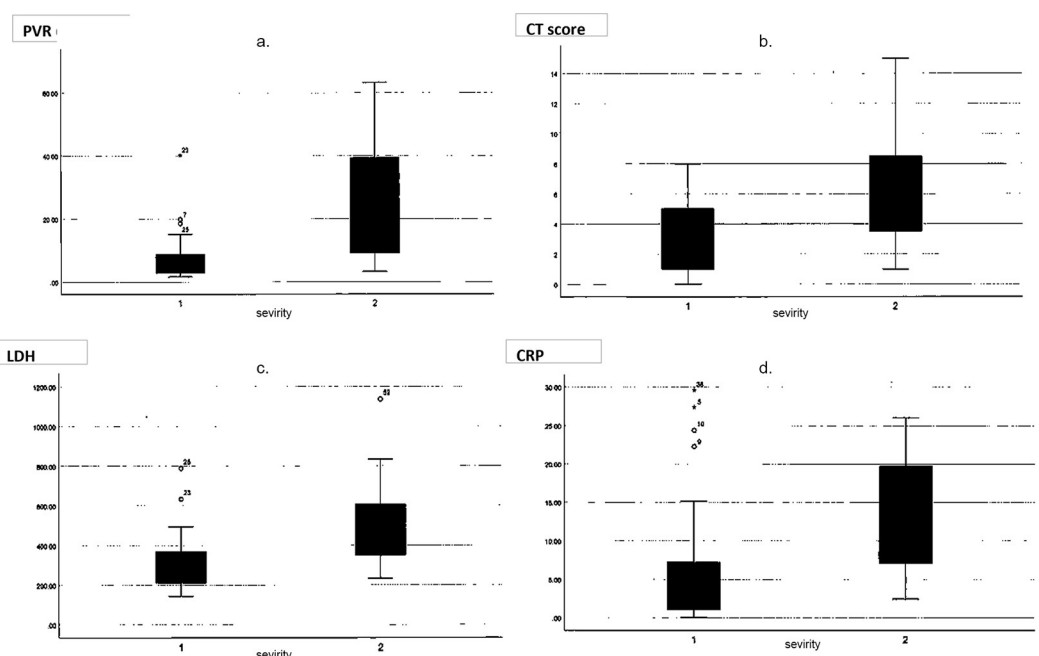

**Fig 6. Boxplots of the PVR, CT score, and CRP.** Boxplots for **a**. The PVR using Z2 ($\geq$–600 HU), **b**. CT scores using Z2 ($\geq$–600 HU), **c**. CRP, and **d**. LDH. 1: Mild group, 2: Severe group. Error bars indicate outliers. CRP, C-reactive protein; CT, computed tomography; LDH, lactate dehydrogenase; PVR, pneumonia volume ratio; ROC, receiver operating characteristic; Z2, Ziostation2.

**Table 3. Cut-off values for pneumonia volume ratio and blood test to differentiate the mild and severe groups.**

| Accuracy of each test | Youden's index | AUC | Sensitivity | Specificity | Lower 95% CI | Upper 95% CI | p-value |
|---|---|---|---|---|---|---|---|
| PVR ($\geq$–600 HU) | 18.69 | 0.881 | 66.7 | 94.7 | 0.781 | 0.981 | <0.001 |
| CT score ($\geq$–600 HU) | 7.5 | 0.77 | 40 | 74 | 0.629 | 0.911 | 0.002 |
| CRP | 5.26 | 0.788 | 86.7 | 68.4 | 0.664 | 0.912 | <0.001 |
| LDH | 306.5 | 0.842 | 86.7 | 68.4 | 0.729 | 0.956 | <0.001 |

AUC, area under the curve; CI, confidence interval; CRP, C-reactive protein; CT, computed tomography; LDH, lactate dehydrogenase; PVR, pneumonia volume ratio.

**Table 4. Bivariable logistic regression of the PVR ($\geq$–600 HU) according to age, sex, number of days from onset to CT, and comorbidities.**

| Predictor | OR (95% CI) | p-value |
|---|---|---|
| PVR ($\geq$–600 HU) | 1.131 (1.048–1.221) | 0.002 |
| Age | 1.031 (0.979–1.086) | 0.246 |
| PVR ($\geq$–600 HU) | 1.124 (1.049–1.206) | 0.001 |
| Sex | 0.000 (0.146–1530.796) | 0.252 |
| PVR ($\geq$–600 HU) | 1.126 (1.044–1.214) | 0.002 |
| Number of days from onset to CT | 1.058 (0.837–1.339) | 0.637 |
| PVR ($\geq$–600 HU) | 1.137 (1.045–1.237) | 0.003 |
| Comorbidities (any) | 9.795 (1.432–67.002) | 0.02 |

CI, confidence interval; CT, computed tomography; OR, odds ratio; PVR, pneumonia volume ratio.

sensitivity and specificity were 66.7% and 89.5% when the PVR threshold was 18, and 60% and 97.4% when the PVR threshold was 20, respectively.

The PVR and CT scores in patients affected by COVID-19-associated pneumonia by Z2 were highly consistent with the visual evaluation results under the condition of $\geq -600$ HU. The AUC and Youden's index of the ROC curve by Z2 ($\geq -600$ HU) were 0.881 and 18.69 for the PVR, and 0.77 and 7.5 for the CT score, respectively, indicating that they are useful for clinical severity classification.

## Discussion

Chest CT plays a major role in COVID-19 treatment, including severity judgment and prognostic prediction. In clinical practice and in previous studies, the spatial progression of pneumonia on CT has been evaluated with the naked eye, and the accuracy and homogeneity have not been ensured.

In this study, we examined the usefulness of determining the severity of COVID-19-associated pneumonia using Z2, an image analysis software widely available in Japan. This methodology can be easily deployed at facilities that have Z2, and thus, has high clinical utility.

Several reports have evaluated the percentage of lesion area of COVID-19-associated pneumonia in each lobe of the lung visually and scored them to determine the disease severity [6–10]. Yang et al. [6] visually classified the percentage of lesion area in each segment as 0%, <50%, and >50%. Li et al. [7] reported that the percentage of lesion area in each lobe was visually classified as 0%, 0–25%, 25–50%, 50–75%, and 75–100%, and scored on a scale of 0–20. The authors found that the optimal threshold for the severe group was 7.5. Francone et al. [9] used a similar classification, with a mortality risk cut-off of 18. Li et al. [8] also reported scores of 0: 0%, 1: <5%, 2: 5–25%, 3: 25–50%, 4: 50–75%, and 5:≥75%, with a cut-off score of 7, a sensitivity of 80%, and a specificity of 82.8% for the severely ill group. The cut-off value for clinical severity classification by CT score varies depending on the method and classification of severity.

The CT scores based on visual evaluations that do not require special software or techniques are widely used in clinical settings. This type of evaluation is subjective; however, it has been reported that the inter-evaluator difference is small, and the results of this study are in agreement. However, the score measurement for each lobe in 25–50% increments is troublesome and imposes a burden on the emergency unit staff. Inoue et al. [11] reported that three visual CT scores evaluations required 25.7–41.7 s, 27.7–39.5 s, and 48.9–80.0 s, respectively. Novel methods for the quantitative and automated measurement of the spatial progression of COVID-19-associated pneumonia have been reported since the early days of the pandemic [12–21].

Using the commercially available image analysis software, Timaran–Montenegro et al. [12] automatically classified −700 to −1000 HU as normal lung and −500 to 20 HU as pneumonia regions, and compared the survival vs. non-survival groups. The percentage of normal lungs was a significant independent factor according to a multinominal logistic analysis. Colombi et al. [13] defined the region of −950 to −700 HU as well as aerated lungs and reported that the measurement by commercially available software and visual measurement were very similar and useful for severity evaluation. In the 10 cases selected in our study, the correlation between the automated measurement by Z2 under the condition of $\geq -600$ HU and the macroscopic measurement was high: very high for the PVR (correlation coefficient 0.842–0.976) and moderate for the CT scores (correlation coefficient 0.651–0.696).

As there were no previous reports of using Z2 as a tool to evaluate diseases, such as pneumonia with increased lung concentration, the concentration range for pneumonia was

determined to be $\geq$-600 HU in this study, based on the high degree of consistency in terms of the visual PVR and CT scores.

The range of normal lung, GGO, and consolidation reported in each study using software varied as follows: –1000 to –600 HU for normal lung, –750 to –100 HU for GGO, and –399 to –69 HU for consolidation [10–15]. Many previous studies set the lower limit of the GGO range at –800 to –700 HU; however, in this study, –600 HU was selected as the lower limit owing to the high degree of agreement with the visual findings. This was probably because it is difficult to recognize a faint increase in concentration based on visual evaluation compared to the software-assisted evaluation. It is an advantage of the software-assisted evaluation that it can detect faint concentrations; however, considering that the CT evaluation of COVID-19-associated pneumonia is generally based on visual evaluation, the detection of faint concentrations that are not measurable by visual evaluation leads to clinical discrepancies.

Grassi et al. [14] reported that the percentage of normal lung, emphysema, and consolidation measured by three different software tools were inconsistent. Granata et al. [15] compared the results obtained from two different software tools and reported that the correlation between them was not high enough. The algorithms, on which each software is based, are different, and therefore, comparisons cannot be made under uniform conditions. Z2 is a software tool owned by more than 300 facilities in Japan. Therefore, an assessment method based on the use of Z2 may be immediately available at these facilities and have a high clinical significance. In addition, the introduction of new technologies is time-consuming and expensive.

Okuma et al. [17] reported that the CT score and the percentage of opacity (PVR in this study) obtained using commercially available artificial intelligence (AI)-based software showed a similar AUC; however, in this study, the AUC corresponding to the PVR and the CT score estimated by Z2 under $\geq$–600 HU was higher in the case of PVR. Theoretically, the CT scoring method can differ by up to 24% in one lobe at the same point, making it less accurate than PVR. Several studies that have evaluated CT findings in cases of COVID-19 pneumonia using software and AI have demonstrated concordance with macroscopic CT scores by converting the PVR into CT scores [11,17,18]. However, as automated measurement of the same standard becomes widespread, the evaluation by PVR is likely to replace CT scores.

Recently, there have been many reports on the diagnosis and severity assessment of COVID-19-associated pneumonia using AI [16–20]. In a study on COVID-19-associated pneumonia using an AI-based software developed by Ziosoft, the company that developed Z2, Aoki et al. [20] measured the CT lesion extent separately for normal lung, GGO, reticulation, and consolidation. In this study, the pneumonia area was evaluated by combining GGO and consolidation; however, more accurate qualitative and quantitative evaluation will be possible if AI-based software is adopted for this purpose in the future.

In this study, Z2 sometimes misidentified subpleural consolidation as extrapulmonary, requiring manual correction. Inoue et al. [11] reported measurement errors with the use of U-NET because of the inclusion of atelectasis, fibrosis, and air trapping in the density mask. When a software tool is used, the measurement is carried out automatically; however, error checking may still need to be performed by human staff.

In this study, we showed the optimal conditions for measuring the PVR and CT score in cases of COVID-19-associated pneumonia using Z2, a widely used image analysis software in Japan, and provided a guideline for clinical severity evaluation based on it. Therefore, defining a Z2-based assessment method has a high clinical significance, and replacing visual evaluation with existing image analysis software represents a way to quickly reduce the burden on clinicians at each facility. We believe that radiologists should consider repurposing existing image analysis software for the evaluation of new viral pneumonia, and it is their responsibility to suggest appropriate methods.

Binomial logistic regression analysis showed no significant effects of age, sex, or time from onset to CT on PVR.

In terms of CT findings, consolidation was significantly higher among the severe group, in agreement with previous reports [9,19–21]. Several laboratory tests have been reported to be indicators of COVID-19 infection. In our study, both CRP and LDH were significant items, again in agreement with previous reports [22,23].

The major limitation of this study was the small number of participants at a single facility. The other limitations were that the manual correction of the subpleural consolidation in the Z2 measurement was performed by a single radiologist and the significance of inter-operator differences was not evaluated. Moreover, PVR assumed the area of $\geq$-600 HU to be a surrogate value for COVID-19 pneumonia, but no histological confirmation was available. The PVR measurements were uniformly performed regardless of the background lesions affecting emphysema, fibrosis, or atelectasis.

In conclusion, we determined the optimal conditions that best approximate visual evaluation for assessing COVID-19-associated pneumonia using Z2, one of the most popular image analysis software tools in Japan, and demonstrated that the AUC of the PVR was higher than that of the CT scores in the assessment of clinical severity. By reusing Z2, faster and more accurate automatic PVR measurements can be made to determine the severity of COVID-19 pneumonia. The introduction of new technologies is time-consuming and expensive; this method has high clinical utility and can be adopted immediately in any facility where Z2 is available for use.

## Supporting information

**S1 Data.**
(PDF)

**S1 File.**
(PDF)

**S2 File.**
(PDF)

**S3 File.**
(PDF)

## Acknowledgments

We would like to thank Dr. Noriko Hida and Dr. Eisuke Inoue for their guidance on the statistical analysis, and Ms. Hokazono and Editage (www.editage.com) for English language editing. We also appreciate the support from our proofreaders and editors. In addition, we are grateful to the clinicians at Fujisawa City Hospital for their insightful advice.

## Author Contributions

**Conceptualization:** Minako Suzuki, Hidefumi Fujisawa.

**Data curation:** Minako Suzuki.

**Formal analysis:** Minako Suzuki.

**Investigation:** Minako Suzuki, Yoshimi Fujii, Yurie Nishimura, Kazuma Yasui.

**Methodology:** Minako Suzuki, Hidefumi Fujisawa.

**Project administration:** Minako Suzuki.

**Resources:** Minako Suzuki.

**Software:** Minako Suzuki.

**Supervision:** Hidefumi Fujisawa.

**Validation:** Minako Suzuki.

**Visualization:** Minako Suzuki.

**Writing – original draft:** Minako Suzuki.

**Writing – review & editing:** Minako Suzuki, Yoshimi Fujii, Hidefumi Fujisawa.

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
