## [Decision Letter · Decision Letter 0]

24 Aug 2023

PONE-D-23-17972Quantitative analysis of chest computed tomography of COVID-19 pneumonia using a software widely used in JapanPLOS ONE

Dear Dr. Suzuki,

Thank you for submitting your manuscript to PLOS ONE. After careful consideration, we feel that it has merit but does not fully meet PLOS ONE’s publication criteria as it currently stands. Therefore, we invite you to submit a revised version of the manuscript that addresses the points raised during the review process.

ACADEMIC EDITOR: Please review comments made by the reviewer and address in the revised manuscript.==============================

We look forward to receiving your revised manuscript.

Kind regards,

Muhammad Adrish, MD, MBA, FCCP, FCCM

Academic Editor

PLOS ONE

Reviewers' comments:

Reviewer's Responses to Questions

**Comments to the Author**

1. Is the manuscript technically sound, and do the data support the conclusions?

Reviewer #1: Yes

Reviewer #2: Partly

2. Has the statistical analysis been performed appropriately and rigorously? 

Reviewer #1: Yes

Reviewer #2: Yes

3. Have the authors made all data underlying the findings in their manuscript fully available?

Reviewer #1: Yes

Reviewer #2: No

4. Is the manuscript presented in an intelligible fashion and written in standard English?

Reviewer #1: Yes

Reviewer #2: Yes

5. Review Comments to the Author

Reviewer #1: The paper was written in a very clear, lucid manner. The topic is interesting and well explained. Sample size is adequate and the results were presented clearly. Figures were clear and labeled. Data is presented well.

Reviewer #2: The authors correctly point out the objective and added value of having such quantitative metrics: one being reducing the time consumption and error in manually tracing the images ( slice by slice) and doing image segmentation. However, their statement pointing to the ineffectualness of the qualitative radiologic report is not fair and is based on comparing apples to oranges with too many confounding variables. There is one main methodological confound embedded in their analyses and a conceptual error of the radiologist’s role in patient care that leads to an erroneous conclusion at the end of their manuscript.

6. PLOS authors have the option to publish the peer review history of their article (what does this mean?). If published, this will include your full peer review and any attached files.

Reviewer #1: No

Reviewer #2: No

---

## [Author Response · Author response to Decision Letter 0]

30 Sep 2023

Response to Reviewer 2

Thank you very much for your comments, which have helped us improve the quality of our work. 

We thank you for acknowledging that the statistical power of our study was adequate and that quantitative studies are useful in predicting clinical outcomes in patients.

 As you pointed out, we did not disclose the research data. We will improve the quality of our work and attach measurement data to the extent that they can be clearly stated.

Moreover, please note that the purpose of this study was not to compare quantitative evaluations made by radiologists and image analysis software (Z2), but to explore the repurposition of the image analysis software (Z2), which is widely used in Japan and was originally intended to evaluate emphysema, for the assessment and potential classification of COVID-19 pneumonia severity.

The Z2 software was released in 2010 and is currently applied in approximately 300 facilities in Japan. 

It can automatically extract the bronchi, lungs, and low-attenuation areas, and can measure the volumes of the entire lung and low-attenuation areas. The area below the threshold density of the lung field is defined as the low attenuation area of the emphysema, and the low attenuation volume is measured for the evaluation of COPD.

In this study, we used Z2 to measure the lung field with a threshold concentration or higher as pneumonia. As Z2 has not been used to evaluate lesions above a certain concentration, such as pneumonia, it was necessary to determine an appropriate threshold in Hounsfield units (HUs). (lines 79–87)

Therefore, for each of the 10 selected cases, an appropriate threshold was determined by comparing the PVR measured manually by two radiologists (lines 118–121), and the PVR was measured with Z2 under different conditions in the range of -500 HU to -700 HU. (lines 135–137)

Undoubtedly, we do not believe that quantitative evaluation alone is sufficient. Two radiologists also performed a qualitative evaluation of CT findings (GGO, crazy-paving, infiltrative shadows, emphysema) in 53 cases (lines 118–119), and the results were consistent with those of previous reports. In particular, there was a significant difference in the clinical severity of the presence of consolidations (lines 331–335). Although not mentioned, both radiologists used both 5-mm slice images and high-resolution CT (HRCT) images for the macroscopic qualitative evaluation of 53 cases.

We believe that determining the appropriate conditions of Z2, originally intended for evaluating emphysema, for evaluating COVID-19 pneumonia is a task that requires the perspective of a radiologist.

In addition, Z2 may not recognize the boundaries between lung field infiltration shadows, atelectasis, and extrapulmonary structures, such as pleural effusions and chest walls; therefore, it is not enough to simply measure automatically by a computer, and therefore, corrections made by a radiologist are needed. (lines 139–144)

We apologize but we did not clearly understand the reviewer 2’s comment on “comparing apples and oranges.” Is this a reference to the comparison between the PVR by Z2 and CT scores by radiologists? If so, there is a misunderstanding and we apologize for the confusion. We compared the PVR with Z2 and CT scores with Z2. The purpose of comparing the two is as follows.

The CT score visually scores each lobe of the lung in 25% increments, and although it is simple and versatile, it is subjective and cumbersome. In contrast, the PVR is automatically measured using software and AI. Thus, objective and accurate results are available quickly.

To measure PVR macroscopically, manual tracing using a slice-by-slice method can be considered, as was done for the 10 cases selected in this study, but this is an effortful and time-consuming process, which is not ideal for clinical practice.

If automatic measurement is possible, it is obvious that it is more accurate to use the PVR rather than the CT score. In this study, the AUC of the ROC for severity classification using PVR and the CT score of Z2 showed that the PVR results exceeded the CT score (lines 247–251).

However, the problem with automatic measurements of PVR is that various software and AI tools measure it using different algorithms. Thus, the results cannot be treated as equivalent. (lines 309–314) Therefore, in fact, past studies on automatic quantitative analysis of chest CT scans of COVID-19 pneumonia have evaluated the PVR by converting it into the CT score (References #11, 17, 18).

In this regard, Z2 is a software that is widely used in Japan; thus, new analysis approaches using Z2 could be easily implemented and would be readily available. (lines 310–314)

We would like to revise the corresponding part in the manuscript to make it easier to understand the benefits of using software that is already on the market rather than using new software or AI.

---

## [Decision Letter · Decision Letter 1]

5 Oct 2023

Quantitative analysis of chest computed tomography of COVID-19 pneumonia using a software widely used in Japan

PONE-D-23-17972R1

Dear Dr. Suzki,

We’re pleased to inform you that your manuscript has been judged scientifically suitable for publication and will be formally accepted for publication once it meets all outstanding technical requirements.

Kind regards,

Muhammad Adrish, MD, MBA, FCCP, FCCM

Academic Editor

PLOS ONE

Additional Editor Comments (optional):

Reviewers' comments:

Reviewer's Responses to Questions

**Comments to the Author**

1. If the authors have adequately addressed your comments raised in a previous round of review and you feel that this manuscript is now acceptable for publication, you may indicate that here to bypass the “Comments to the Author” section, enter your conflict of interest statement in the “Confidential to Editor” section, and submit your "Accept" recommendation.

Reviewer #2: All comments have been addressed

2. Is the manuscript technically sound, and do the data support the conclusions?

Reviewer #2: Yes

3. Has the statistical analysis been performed appropriately and rigorously? 

Reviewer #2: Yes

4. Have the authors made all data underlying the findings in their manuscript fully available?

Reviewer #2: No

5. Is the manuscript presented in an intelligible fashion and written in standard English?

Reviewer #2: Yes

6. Review Comments to the Author

Reviewer #2: The authors have appropriately addressed the concerns raised during the previous review and I think that the manuscript is ready for publication.

7. PLOS authors have the option to publish the peer review history of their article (what does this mean?). If published, this will include your full peer review and any attached files.

Reviewer #2: No

---

## [Editor Report · Acceptance letter]

9 Oct 2023

PONE-D-23-17972R1 

Quantitative analysis of chest computed tomography of COVID-19 pneumonia using a software widely used in Japan 

Dear Dr. Suzuki:

I'm pleased to inform you that your manuscript has been deemed suitable for publication in PLOS ONE. Congratulations! Your manuscript is now with our production department. 

Kind regards, 

on behalf of

Dr. Muhammad Adrish 

Academic Editor

PLOS ONE